# Antinociceptive and anti-inflammatory effects of hydrazone derivatives and their possible mechanism of action in mice

**Maria Alice Miranda Bezerra Medeiros**[1,2], **Mariana Gama e Silva**[1,3], **Jackson de Menezes Barbosa**[1,3], **Érica Martins de Lavor**[1,2], **Tiago Feitosa Ribeiro**[3], **Cícero André Ferreira Macedo**[2], **Luiz Antonio Miranda de Souza Duarte-Filho**[4], **Thiala Alves Feitosa**[1,4], **Jussara de Jesus Silva**[5], **Harold Hilarion Fokoue**[6], **Cleônia Roberta Melo Araújo**[5], **Arlan de Assis Gonsalves**[5], **Luciano Augusto de Araújo Ribeiro**[4,5], **Jackson Roberto Guedes da Silva Almeida**[1,2]*

1 Núcleo de Estudos e Pesquisas de Plantas Medicinais (NEPLAME), Universidade Federal do Vale do São Francisco, Petrolina, Pernambuco, Brasil, 2 Pós-Graduação em Biotecnologia, Universidade Estadual de Feira de Santana, Feira de Santana, Bahia, Brasil, 3 Pós-Graduação em Biotecnologia—Rede Nordeste de Biotecnologia (RENORBIO), Universidade Federal Rural de Pernambuco, Recife, Pernambuco, Brasil, 4 Pós Graduação em Biociências, Universidade Federal do Vale do São Francisco, Petrolina, Pernambuco, Brasil, 5 Colegiado de Farmácia, Universidade Federal do Vale do São Francisco, Petrolina, Pernambuco, Brasil, 6 Laboratório de Avaliação e Síntese de Substâncias Bioativas (LASSBio), Universidade Federal do Rio de Janeiro, Rio de Janeiro, Rio de Janeiro, Brasil

* jackson.guedes@univasf.edu.br

**Data Availability Statement:** All relevant data are within the paper and its Supporting Information files.

## Abstract

Pain and inflammation are unpleasant experiences that usually occur as a result of tissue damage. Despite the number of existing analgesic drugs, side effects limit their use, stimulating the search for new therapeutic agents. In this sense, five hydrazone derivatives (H1, H2, H3, H4, and H5), with general structure $R_1R_2C = NNR_3R_4$, were synthesized with molecular modification strategies. In this paper, we describe the ability of hydrazone derivatives to attenuate nociceptive behavior and the inflammatory response in mice. Antinociceptive activity was evaluated through acetic acid-induced writhing and formalin-induced nociception tests. In both experimental models, the hydrazone with the greatest potency (H5) significantly ($p < 0.05$) reduced nociceptive behavior. Additionally, methods of acute and chronic inflammation induced by different chemicals (carrageenan and histamine) were performed to evaluate the anti-inflammatory effect of H5. Moreover, molecular docking analysis revealed that H5 can block the COX-2 enzyme, reducing arachidonic acid metabolism and consequently decreasing the production of prostaglandins, which are important inflammatory mediators. H5 also changes locomotor activity. In summary, H5 exhibited relevant antinociceptive and anti-inflammatory potential and acted on several targets, making it a candidate for a new multi-target oral anti-inflammatory drug.

## Introduction

Pain is an unpleasant sensory and emotional experience that usually occurs as a result of tissue damage. This important public health problem causes disability, suffering, and is associated

**Funding:** The authors would like to thank Prof. Lídia Moreira Lima and Eliezer J. Barreiro, and the Instituto Nacional de Ciência e Tecnologia em Fármacos e Medicamentos (INCTINOFAR), Project CNPq 465.249/2014-0, Project FAPERJ E-26/ 010.000090/2018, for the financial support.

**Competing interests:** The authors have declared that no competing interests exist.

with increased anxiety [1]. Pharmacological treatment of pain initially includes non-opioids, followed by opioids, and finally, if necessary, adjuvants like anticonvulsants and antidepressants. However, despite the number of available analgesic drugs, side effects limit their use, stimulating the search for new therapeutic agents [2].

Hydrazone derivatives are a class of organic compounds with the general structure $R_1R_2C = NNR_3R_4$ [3], being considered Schiff bases. Normally, hydrazones are substances obtained by the condensation of hydrazines with ketones or aldehydes, being products of the classic derivatization of carbonyl compounds [4, 5].

In this work, we used hydralazine hydrochloride as the organic hydrazine for the synthesis of the desired hydrazones with a novel chemical structure. The use of this drug for the synthesis of new organic compounds and studies of medicinal chemistry has been increasing in scientific publications year by year. It has been reported that hydrazine has a privileged chemical structure for the coordination with metallic cations [5], and for its biological activity, mainly due to the presence of the pyridazine heterocycle in its structure [6].

Hydrazones and their derivatives are known to exhibit a wide range of interesting biological activities like antioxidant, analgesic, antimicrobial, anticancer and also can act as potent anti-inflammatory agents [7]. In the last two decades, preclinical studies testing hydrazone derivatives in different models have been extensively reported in the scientific literature [8]. Such panorama indicates that the application of preclinical tests, including *in vivo* experiments, has an important role in drug discovery. For this reason, we decided to evaluate novel synthesized hydrazone derivatives regarding their antinociceptive and anti-inflammatory potential.

That being said, five hydrazone derivatives (compounds H1, H2, H3, H4, and H5) were prepared based on the combination of hydralazine with previously synthesized α,β-unsaturated carbonyl compounds. Studies have shown that the combination of hydrazones with other functional groups improves its biological properties and provides pharmacologically active molecules [9]. Additionally, a variety of hydrazone derivatives has been developed to minimize the gastrointestinal discomfort and toxicity commonly related to analgesic drugs [10].

Thus, this work evaluates the antinociceptive activity of the hydrazone derivatives in different experimental models as well as its possible mechanism of action in mice. Moreover, we performed docking studies with some of the main targets responsible for nociceptive and inflammatory processes, in order to understand their interactions on the molecular level.

## Materials and methods

Using the methodologies described by Murtinho and coworkers, the synthetic intermediates i1-i5 (α,β-unsaturated ketones) were prepared using aldolic condensation reaction between aldehydes (a1-a5) and propanone [11]. Hydrazone derivatives (H1-H5) were prepared using condensation reaction between carbonyl compounds (i1-i5) and inorganic hydrazine (hydralazine drug), and using the methodologies described by Ananthnag and coworkers [12]. Fig 1 shows the synthetic route used for the preparation of these hydrazone derivatives.

### Animals and ethics statement

We conducted all experiments using 8-week-old male Swiss mice (*Mus musculus*) (30–40 g), totalizing 378 animals. For all experiments, each mouse participated in a single painful protocol so that none of them were reused. Mice were kept in groups of six individuals (n = 6) in polypropylene cages at room temperature set at $22 \pm 1\degree$C with a relative humidity of 60–80%, light/dark cycle of 12 h (start 06:00 and end 18:00), and water and food (Purina Labina) *ad libitum*. This work was developed according to the Conselho Nacional para o Controle de Experimentação Animal (CONCEA, Brazil) and implemented following the recommendations of the

**Fig 1. Synthetic route for hydrazone derivatives H1-H5.**

International Association for the Study of Pain [13, 14]. All experimental procedures were accredited by the Comitê de Ética no Uso de Animais of the Universidade Federal do Vale do São Francisco (CEUA-UNIVASF, Brazil) under the authorization number 0004/241017. We did all possible to mitigate animal suffering. After each protocol, mice were anesthetized with an injection of 60 mg/kg of ketamine associated with 7.5 mg/kg of xylazine, intraperitoneally, followed by cervical dislocation. Syringes of 1 ml with a needle of 13 x 0.45 mm were used for intraperitoneal injections whereas a gavage needle was used for the oral route administrations [14].

### Acetic acid-induced writhing test

For the initial screenings, we chose the writhing test as a model to evaluate the analgesic effect of the hydrazone derivatives [14, 15]. The referred test was performed as described by Collier and collaborators [14, 16] with minor adjustments. To perform it, mice were split into thirteen groups of six animals each (n = 78), being treated orally (p.o.) with H1, H2, H3, H4, H5 (20 and 40 mg/kg, p.o.) or saline (negative control, p.o.) 1 h before the nociceptive agent (10 ml/kg of a 0.9% acetic acid solution) [14, 17]. After five minutes of the acetic acid injection, the number of abdominal writhing was recorded for 10 min [14, 18]. Indomethacin (20 mg/kg, i.p.) and morphine (10 mg/kg, i.p.) were used as reference drugs for anti-inflammatory and antinociceptive activities, respectively, being administered 30 min before the nociceptive agent. Lastly, writhing behavior was defined as the contractions of the abdominal muscles with pelvic rotation, followed by hind limb extension [14].

## Formalin-induced nociception test

We executed the formalin test as Hunskaar and Hole described [14, 19]. To do so, mice were split into thirteen groups of six individuals (n = 78). One hour before formalin injection, mice were pretreated with saline (p.o.), H1, H2, H3, H4, or H5 (20 and 40 mg/kg, p.o). The reference drugs indomethacin (20 mg/kg, i.p.) and morphine (10 mg/kg, i.p.) were given half an hour prior to the formalin injection. The 2.5% (v/v) formalin solution was prepared in 0.9% sterile saline (20 μl/animal) and injected into the right hind paw of mice [14, 20]. Right after formalin injection, mice were returned to the mirror chambers, being observed for 30 min. The total time (in seconds) that the animal spent licking and/or biting its injected paw was used as a pain indicator. Typically, formalin injection elicits a biphasic nociceptive response: (I) an acute phase (5 min after formalin injection) with a quiescent phase of approximately 10 min and (II) a longer-lasting tonic phase (15 to 30 min after formalin injection) [14, 21]. In another set of experiments, mice were split into twenty-one groups of six individuals (n = 126). In these experiments, we assessed the participation of the ATP-sensitive potassium channels and the vanilloid, muscarinic, opioid, nitrergic, and serotonergic systems in the antinociceptive effect of H5, the most promising hydrazone. To do so, thirty minutes before treatment with H5 (20 mg/kg p.o.), mice were administered with the respective blockers: glibenclamide (2 mg/kg, i.p.), ruthenium red (3 mg/kg, i.p.), atropine (0.1 mg/kg, i.p.), naloxone (1.5 mg/kg, i.p.), *N*(G)-Nitro-*L*-arginine methyl ester (*L*-NAME, 20 mg/kg, i.p), and ondansetron (0.5 mg/kg, i.p) [14, 22]. Then, the total licking and/or biting time was measured as described above.

## Leukocyte migration to the peritoneal cavity induced by carrageenan

The induction of leukocyte migration was performed by injecting 250 μl of a 1% carrageenan solution (i.p.) into the peritoneal cavity of mice. This procedure was performed one hour after saline (p.o.) or H5 (20 and 40 mg/kg, p.o.) administration and half an hour after dexamethasone injection (2 mg/kg, i.p.). Four hours later, mice (n = 24) were euthanized as described above. Their peritoneal cavity was washed with 3 ml of a 1 mM EDTA solution (in saline) [23]. Then, the washing fluid was collected and centrifuged (3000 rpm for 6 min) at room temperature. Subsequently, an aliquot of 10 μl of the centrifuged suspension was mixed with 200 μl of Turk solution. To count the total number of cells, a Neubauer chamber was used. These results are expressed as the number of leukocytes per milliliter (leukocyte/ml) [14, 24].

## Carrageenan-induced hind paw edema

In this experiment, we divided mice into five groups of six animals (n = 30). One hour before subcutaneous injection of carrageenan (2.0% carrageenan) or saline (0.9%) into their right hind paw (20 μl/animal), mice were pretreated with H5 (20 and 40 mg/kg, p.o), saline (p.o.), or indomethacin (20 mg/kg, i.p.) [14, 25, 26]. After, using a plethysmometer (PanLab LE 7500, Spain), the mice paw volume (from nails up to the ankle joint) was measured 0, 1, 2, 3, and 4 h after injection of carrageenan, as described earlier [14, 27]. The calculations of the inhibition of the paw edema were performed according to the following formula:

$$edema = \frac{(paw\ volume\ at\ measurement\ time - initial\ paw\ volume)}{initial\ paw\ volume}$$

## Histamine-induced hind paw edema

To assess the participation of histaminic receptors, mice were split into three groups of six individuals (n = 18). One hour before histamine injection, the animals were pretreated with H5 (20 mg/kg, p.o), saline (p.o.), or indomethacin (20 mg/kg, i.p.). Then, histamine (100 μg/

paw) or saline (0.9%) were administered subcutaneously into mice's right hind paw at a volume of 20 μl/animal [14, 28]. After subcutaneous injections, the paw volume was measured at minutes 0, 30, 60, 90, 120, and 150 [14, 29].

### Rota-rod test

A rota-rod apparatus (Insight, Brazil) was used for the assessment of motor coordination. Initially, animals capable to stand on the rota-rod apparatus for 60 s (7 rpm) were selected 24 h before the test. The mice were divided into four groups of six animals each (n = 24) and were pretreated with H5 (20 and 40 mg/kg, p.o), saline, or diazepam (2.5 mg/kg). Each animal was individually evaluated on the rota-rod apparatus at 30, 60, 90, and 120 min after treatments, and the time (s) spent on top of the bar was recorded for up to 180 s [30, 31].

### *Artemia salina* toxicity test

The methodology described by Meyer et al. (1982) with some modifications was performed to evaluate the toxicity of hydrazone H5 against brine shrimps (*Artemia sp*. nauplii). *Artemia salina* cysts (20 mg) were incubated in 1000 ml of saltwater (38 g/l) under illumination. After hatching, a series of concentrations ranging from 1 to 1000 μg/ml of hydrazone H5 in 5 ml of saline water containing 10 *Artemia sp*. (triplicate) were obtained. The nauplii were exposed to solutions for 24 h and 48 h, when the mortality was accounted for calculation of the lethal concentration 50% ($LC_{50}$) [32].

### Physicochemical properties and ADMET profile

We used *in silico* predictions to assess the physicochemical properties and ADMET profile of hydrazones, using the ACD/Percepta Program. The anti-inflammatory drugs indomethacin and meloxicam were also used for comparison purposes.

### Molecular docking analysis

Molecules were constructed on Spartan'16 (Wavefunction Inc.) software and conformational analysis by molecular mechanic method (MMFF—Merck molecular force field) was performed. Starting from minor energy conformer, equilibrium geometry was calculated by PM6 semi empiric method [33].

We obtained the X-ray crystallographic structure of murine COX-2 enzyme, complexed with meloxicam (MXM), from the RCSB Protein Data Bank (PDB ID: 4M11) [34, 35].

Molecular docking studies were performed in triplicate with GOLD 5.4 with all scoring functions available (ChemPLP, GoldScore, ChemScore, and ASP) [36], with the default parameter. The binding site was determined within 8Å around the MXM as a reference. 10 poses were generated for each compound and the best scoring complexes for each ligand were selected. Firstly, for validation purposes, meloxicam (MXM) was redocked in the binding site to evaluate the accuracy of the docking procedure with the 4 function, in this system, evaluating the RMSD (Root Mean Square Deviation) between the native and post-redocking conformation of MXM. In order to check H5 possible interaction modes and score value, the aforementioned procedure was performed.

GOLD uses a genetic algorithm for docking compounds into protein (3D crystallographic structure or 3D model) binding sites [37]. GOLD presents high accuracy and reliability and considers the full ligand flexibility and partial protein flexibility.

### Statistical analysis

The results are presented as the mean ± standard error of the mean (SEM), and statistical analysis was performed using one-way analysis of variance (ANOVA) followed by Tukey's test. Values of $p < 0.05$ were considered statistically significant. All analyses were performed using GraphPad Prism 5.0 (Graph Pad Prism Software, Inc., San Diego, CA, USA).

## Results and discussion

Firstly, the antinociceptive effect of hydrazones derivatives (H1-H5) was evaluated using the acetic acid-induced nociception test. In this protocol, H1 attenuated the nociceptive activity, reducing the number of writhings by 83.87% and 78.78% at the both tested doses (20 mg/kg and 40 mg/kg), as shown in Fig 2A. H2 reduced the nociceptive effect of acetic acid, reducing the number of writhings by 96.00% and 89.93% at both tested doses (20 mg/kg and 40 mg/kg),

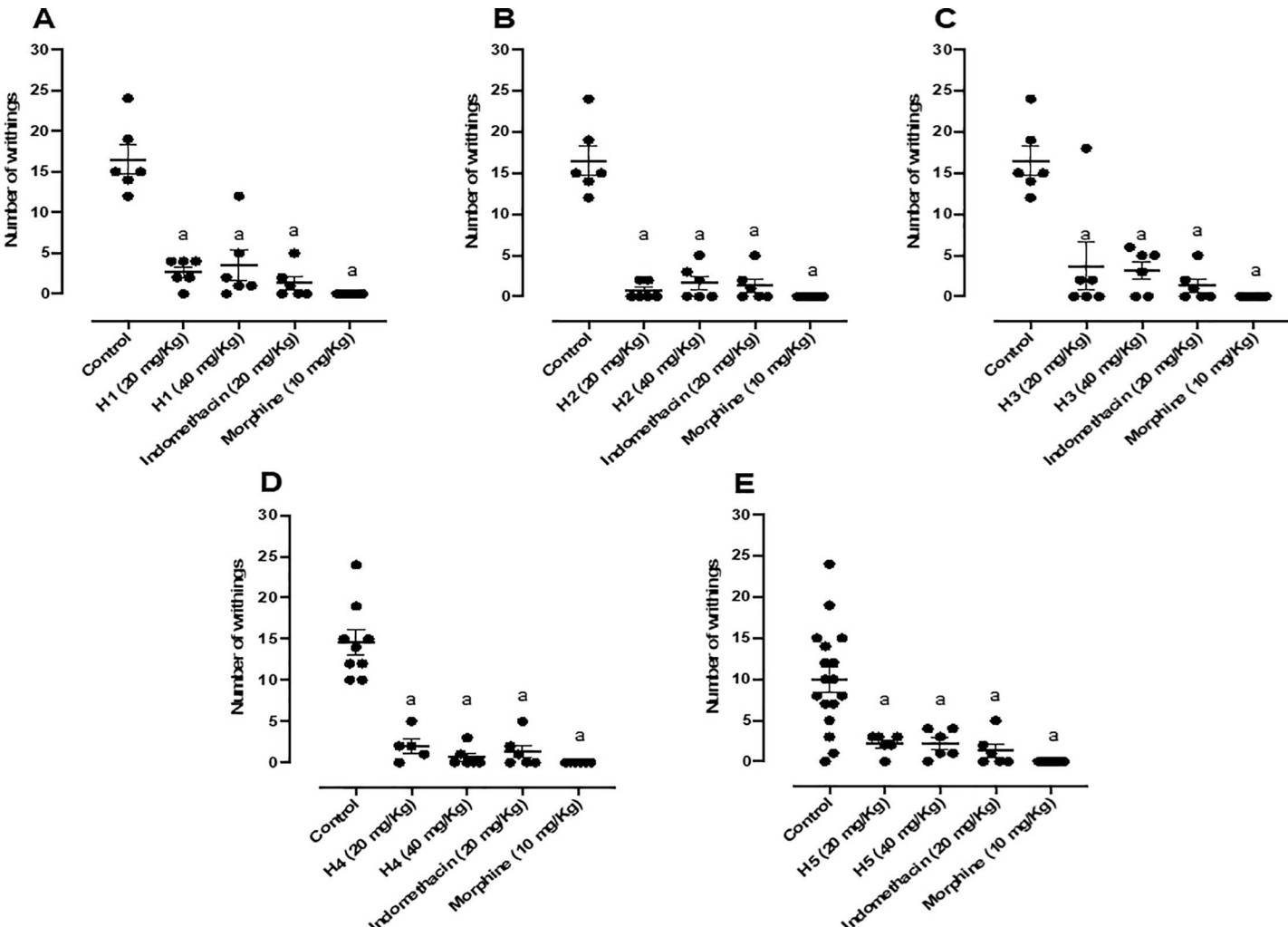

**Fig 2.** (A) Effect of H1 (20 and 40 mg/kg, p.o.), morphine (10 mg/kg, i.p.) and indomethacin (20 mg/kg, i.p.); (B) Effect of H2 (20 and 40 mg/kg, p.o.), morphine (10 mg/kg, i.p.), and indomethacin (20 mg/kg, i.p.); (C) Effect of H3 (20 and 40 mg/kg, p.o.), morphine (10 mg/kg, i.p.) and indomethacin (20 mg/kg, i.p.); (D) Effect of H4 (20 and 40 mg/kg, p.o.), morphine (10 mg/kg, i.p.) and indomethacin (20 mg/kg, i.p.); (E) Effect of H5 (20 and 40 mg/kg, p.o.), morphine (10 mg/kg, i.p.) and indomethacin (20 mg/kg, i.p.) in the acetic acid-induced writhing test in mice (n = 6, per group). Values are expressed as the mean ± SEM, where *a* indicates $p < 0.05$, significantly different from the control group, according to ANOVA, followed by Tukey's test.

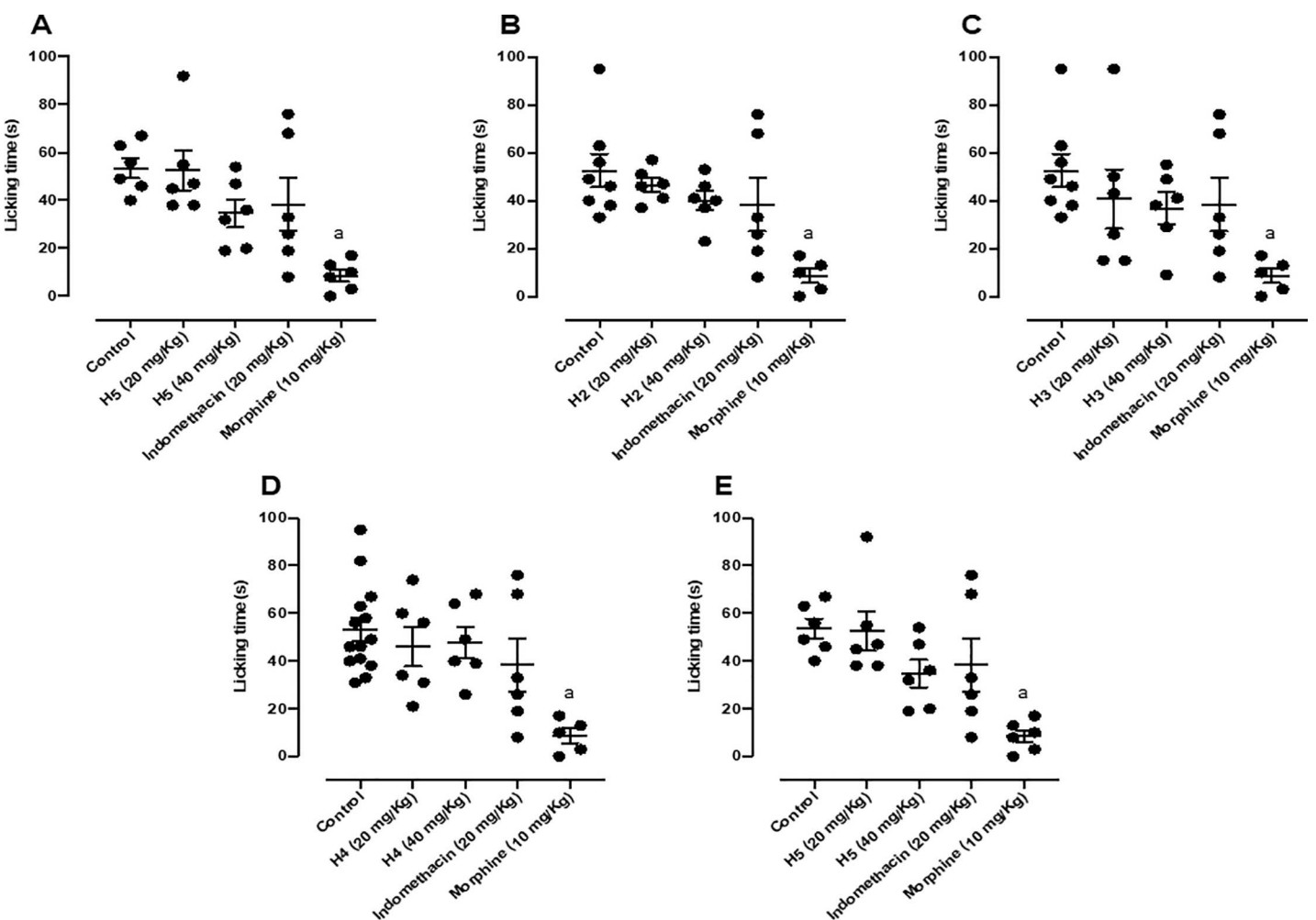

**Fig 3.** (A) Effect of H1 (20 and 40 mg/kg, p.o.), morphine (10 mg/kg, i.p.) and indomethacin (20 mg/kg, i.p.); (B) Effect of H2 (20 and 40 mg/kg, p.o.), morphine (10 mg/kg, i.p.) and indomethacin (20 mg/kg, i.p.); (C) Effect of H3 (20 and 40 mg/kg, p.o.), morphine (10 mg/kg, i.p.) and indomethacin (20 mg/kg, i.p.); (D) Effect of H4 (20 and 40 mg/kg, p.o.), morphine (10 mg/kg, i.p.) and indomethacin (20 mg/kg, i.p.); (E) Effect of H5 (20 and 40 mg/kg, p.o.), morphine (10 mg/kg, i.p.) and indomethacin (20 mg/kg, i.p.) in the first phase of the formalin-induced nociception test in mice (n = 6, per group). Values are expressed as the mean ± SEM, where *a* indicates $p < 0.05$, significantly different from the control group, followed by Tukey's test.

as shown in Fig 2B. H3 also reduced the nociceptive effect of acetic acid, reducing the number of writhings by 77.81% and 80.84% at both tested doses (20 mg/kg and 40 mg/kg), as shown in Fig 2C. H4 attenuated the acetic acid-induced nociceptive effect, reducing the number of writhings by 87.87% and 96.00% at both tested doses (20 mg/kg and 40 mg/kg), as shown in Fig 2D. Finally, H5 was also able to mitigate the nociceptive effect promoted by acetic acid, reducing the number of writhings by 86.90% at both tested doses (20 mg/kg and 40 mg/kg), as shown in Fig 2E. Indomethacin and morphine results were the same for the Fig 2A–2E, as the experiments were carried out on the same day, the animals used had the same age and weight range, in addition to being exposed to the same conditions.

According to the statistical analysis, all hydrazones tested had similar effects compared to indomethacin, which inhibited 91.93% of nociception. These results were also similar to *N*-acylhydrazone LASSBio 1586, which presented 88.97% of inhibition at the highest dose of 40 mg/kg [14].

The acetic acid-induced writhing test is very nonspecific. The intraperitoneal injection of this nociceptive agent causes the activation of nociceptors, inducing the release of a variety of pain mediators, such as histamine, bradykinin, serotonin, glutamate, noradrenaline, substance P, nitric oxide, and prostaglandins [14, 16]. Because of this, it is not possible to determine the specific pharmacological pathways involved in the effect of a substance. In this sense, we performed the formalin-induced nociception test.

This test can shed light on the two phases of the nociceptive response. The first one corresponds to a nociceptive response triggered by mediators that act primarily in the central nervous system, through the activation of serotonergic, muscarinic, vanilloid, and glutamatergic receptors. In the second phase, it is the inflammatory mediators' histamine, bradykinin, and prostaglandins that take place and participate in the nociceptive response [14, 19].

All five hydrazone derivatives mitigated the nociceptive response elicited by formalin. When mice were pretreated with H1 and submitted to the formalin test, there was a reduction in nociceptive behavior only in the second phase of nociception, with the doses of 20 and 40 mg/kg being responsible for 59.61% and 39.64% of the normal antinociceptive effect, respectively (Fig 4A). H2 attenuated nociceptive behavior only in the second phase of nociception, being the dose of 20 mg/kg responsible for 51.67% of normal antinociceptive effect (Fig 4B). H3 reduced nociceptive behavior only in the second phase of nociception, being the dose of 40 mg/kg responsible for 64.04% of normal antinociceptive effect (Fig 4C). H4 reduced nociceptive behavior only in the second phase of nociception too, being the doses of 20 and 40 mg/kg responsible for 96.41% and 78.16% of normal antinociceptive effect respectively (Fig 4D). Lastly, H5 diminished nociceptive behavior only in the second phase of nociception, being the doses of 20 and 40 mg/kg responsible for 78.92% and 100% of normal antinociceptive effect respectively (Fig 4E). Indomethacin and morphine results were the same for the Fig 3A–3E, and indomethacin and morphine results were the same for the Fig 4A–4E, as the experiments were carried out on the same day, the animals used had the same age and weight range, in addition to being exposed to the same conditions.

Based on these results, the pharmacologically and chemically similar hydrazones H4 and H5 showed greater antinociceptive potency. To perform the remaining pharmacological tests, however, we decided to use H5. Regarding the statistical analysis, there was no statistical difference between the two doses. Similar result were observed for indomethacin (97.25%) and LASSBio 1586 (96.74%) at a dose of 20 mg/kg [14]. According to these results, H5 does not affect the first phase of the test, suggesting that this chemical has a peripheral (not central) effect that reduces nociception only in the second phase of the formalin test, similar to indomethacin.

Similarly to our results with H5, Meymandi *et al.* (2019) showed that celecoxib (10–30 mg/kg), a specific COX-2 inhibitor, had antinociceptive and anti-inflammatory activity in mice submitted to the formalin test, being this effect visible only in the second phase of the test [38–40]. This effect similar to celecoxib is an important pharmacological indicator since we are proposing H5 as an antinociceptive drug candidate. Therefore, it is plausible to state that H5 may have an inhibitory effect on the COX-2 enzyme.

Mice treated with H5 had a reduction in nociception and inflammation. In this context, we conducted several tests to explore the antinociceptive and anti-inflammatory mechanisms involved in such effects.

When animals were pretreated with naloxone (1.5 mg/kg, i.p.), the pharmacological effect of H5 (20 mg/kg, p.o.) was completely reversed in the second phase of the test (Fig 5), suggesting that its peripheral antinociceptive response was involved at least in part with the opioid system.

The antinociceptive effect of H5 was fully reversed in the second phase of the formalin test when animals were pretreated with naloxone. This suggests that its peripheral antinociceptive effect depends, at least in part, on the opioid system. Naloxone is an opioid antagonist and for

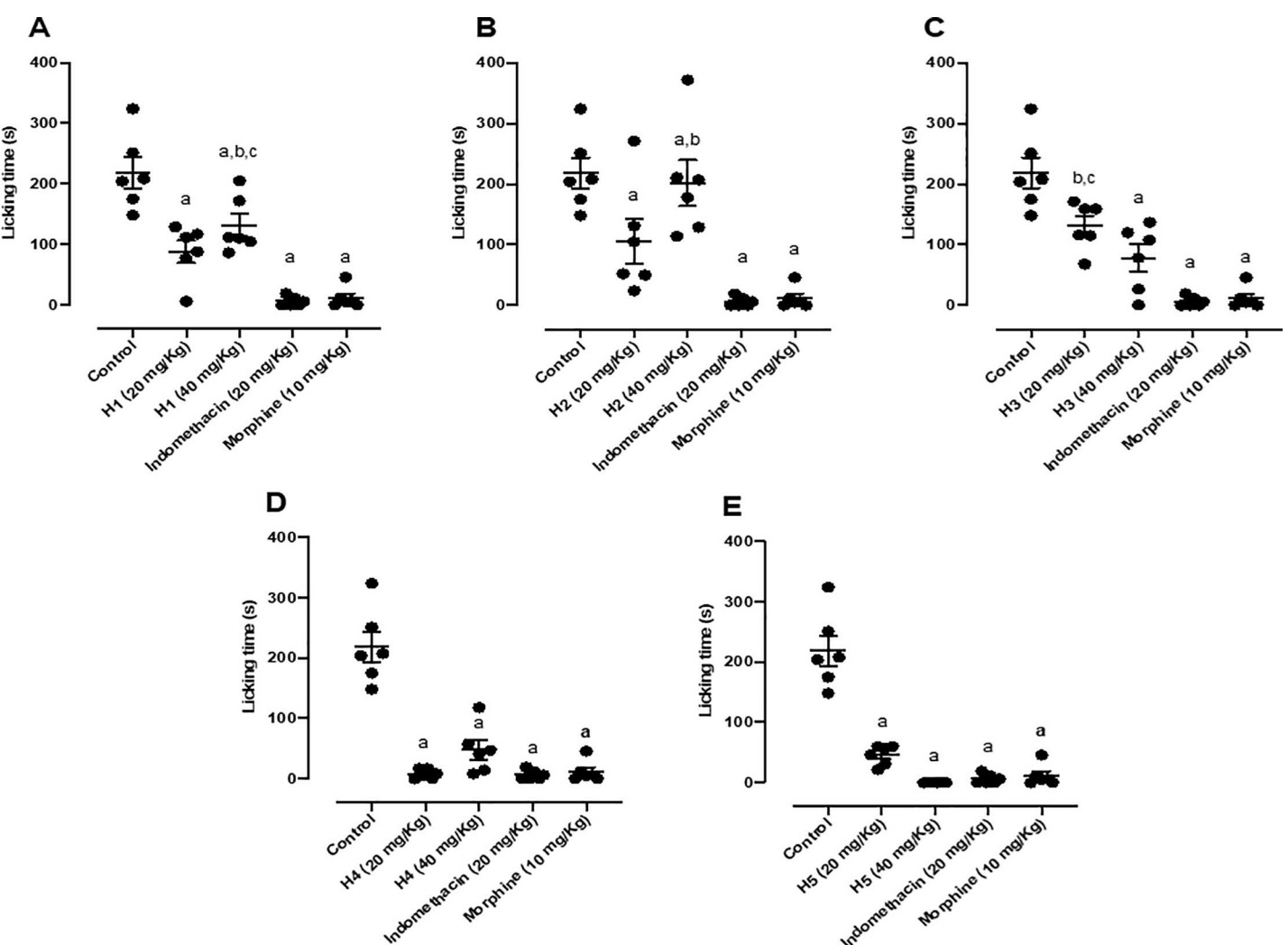

**Fig 4.** (A) Effect of H1 (20 and 40 mg/kg, p.o.), morphine (10 mg/kg, i.p.) and indomethacin (20 mg/kg, i.p.); (B) Effect of H2 (20 and 40 mg/kg, p.o.), morphine (10 mg/kg, i.p.) and indomethacin (20 mg/kg, i.p.); (C) Effect of H3 (20 and 40 mg/kg, p.o.), morphine (10 mg/kg, i.p.) and indomethacin (20 mg/kg, i.p.); (D) Effect of H4 (20 and 40 mg/kg, p.o.), morphine (10 mg/kg, i.p.) and indomethacin (20 mg/kg, i.p.); (E) Effect of H5 (20 and 40 mg/kg, p.o.), morphine (10 mg/kg, i.p.) and indomethacin (20 mg/kg, i.p.) in the second phase of the formalin-induced nociception test in mice (n = 6, per group). Values are expressed as the mean ± SEM, where *a* indicates $p < 0.05$, significantly different from the control group, *b* indicates $p < 0.05$ in comparison with indomethacin group and *c* indicates $p < 0.05$ in comparison with morphine group, according to ANOVA, followed by Tukey's test.

this reason, it significantly blocks the activity of morphine in both phases of the formalin test. The results of Mehanna *et al.* (2018) [41] demonstrated that naloxone completely reversed the effect of tadalafil in the first phase of the same test and partially in the second phase, suggesting that this drug have a peripheral antinociceptive effect that activates the opioid receptors, which was also demonstrated by Florentino *et al.* (2015) with pyrazole compounds [41–44].

In agreement with our data, we found in the scientific literature that the antinociceptive effect of a given agent may involve peripheral opioid receptors. An example of that is the bergamot essential oil (BEO)-induced antinociception, which, according to Komatsu and colleagues, has its antinociceptive effect related to the peripheral activation of µ and κ-opioid receptors [45]. Beyond that, flavonoids also have the peripheral ability to decrease hyperalgesia since this effect depends on the activation of µ and δ-opioid receptors located outside of the brain

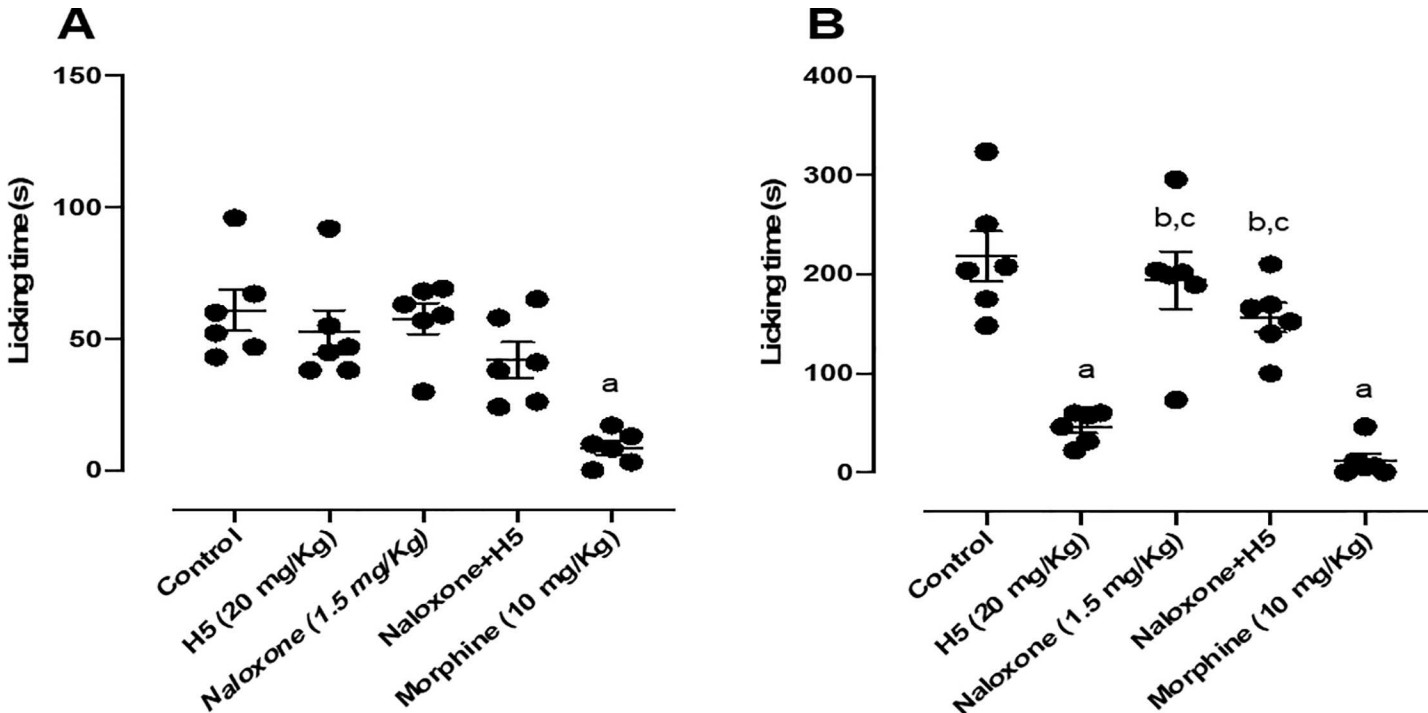

**Fig 5.** Effect of H5 (20 mg/kg, p.o.), naloxone (1.5 mg/kg, i.p.), naloxone + H5 and morphine (10 mg/kg, i.p.) in the first (A) and second (B) phases of the formalin-induced nociception test in mice (n = 6, per group). Values are expressed as the mean ± S.E.M., where *a* indicates *p* < 0.05 in comparison with control group, *b* indicates *p*<0.05 in comparison with morphine group, and *c* indicates *p* < 0.05 in comparison with H5 group, according to ANOVA, followed by Tukey's post-test.

[46–48]. In addition, several studies have shown that activation of peripheral opioid receptors inhibits inflammatory pain and activates *L*-arginine/NO/cGMP pathway [41, 43, 44].

When animals were pretreated with *L*-NAME (20 mg/kg, i.p.) the pharmacological effect of H5 (20 mg/kg, p.o.) was not reversed in the second phase of the test (Fig 6), suggesting that its peripheral antinociceptive response was not involved with the nitrergic system.

When nociceptors are activated, intracellular signaling cascades lead to an increase in the production of a variety of neuromodulators such as NO and cGMP. Thus, a sufficient increase of NO concentration boosts the cGMP production and leads to the activation of glutamatergic receptors. These receptors are known to mediate painful sensations, making the NO concentrations directly associated with nociception [49–53]. The cGMP acts directly or through the stimulation of protein kinases that phosphorylate ion channels, favoring the firing of action potentials that culminate in the production of nociception [53–55]. Systemically, the *L*-arginine/NO/cGMP pathway blockade causes a decrease in nociception [53, 56]. In the second phase of the formalin test, in the presence of *L*-NAME, H5 showed no reversibility of its antinociceptive effect. This result corroborates the one presented by Silva et al. 2018 about the LASSBio 1586 [14].

When the animals were pretreated with ondansetron (0.5 mg/kg, i.p.), H5 (20 mg/kg, p.o.) did not have its pharmacological effect affected in the second phase of the test (Fig 7), suggesting that its peripheral antinociceptive response do not depend on the serotonergic system.

Ondansetron did not alter the antinociceptive effect of H5. Therefore, its effect is not involved with the serotonergic system. Similar to LASSBio 1586 in the second phase, H5 also showed no reversibility of its antinociceptive effect [14]. Diverse serotonin (5-HT) receptors are present in the central and peripheral nervous systems [57, 58]. Studies have shown that

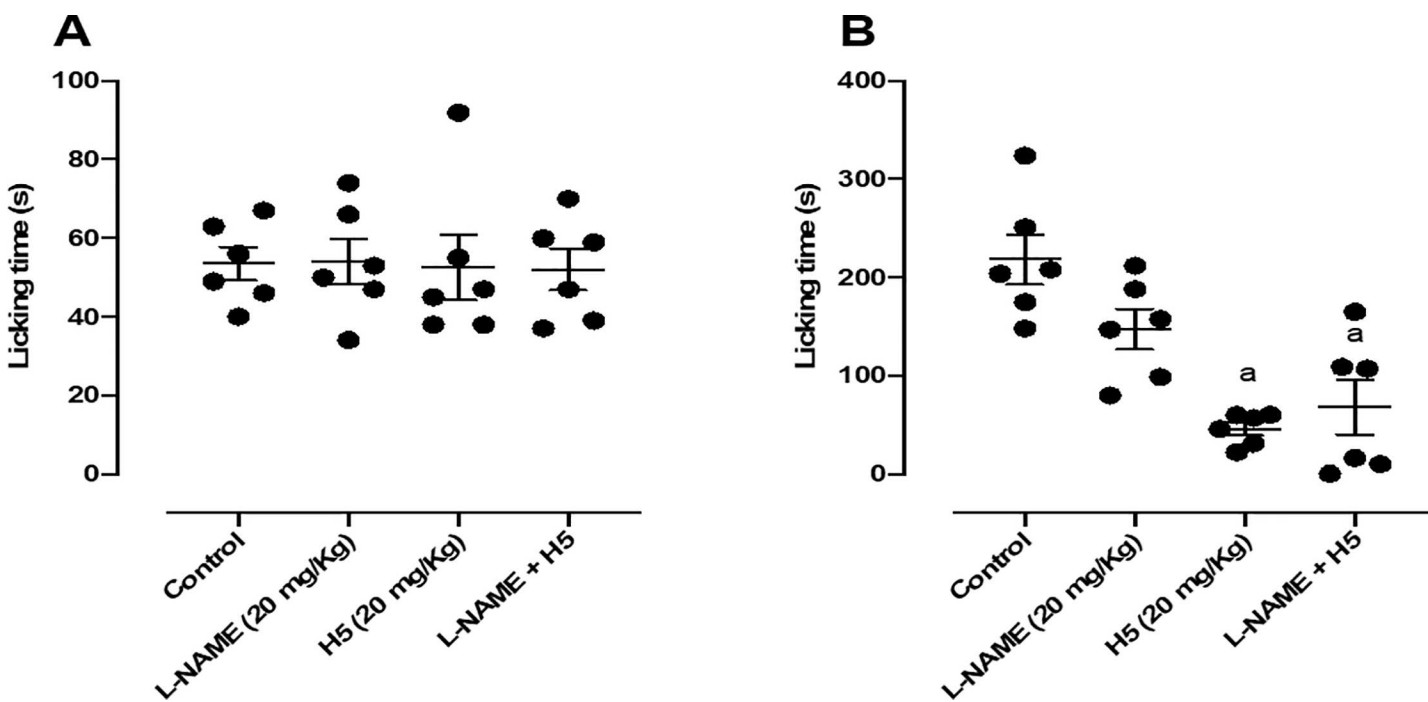

**Fig 6.** Effect of *L*-NAME (20 mg/kg, i.p.), H5 (20 mg/kg, p.o.), *L*-NAME + H5, in the first (A) and second (B) phases of the formalin-induced nociception test in mice (n = 6, per group). Values are expressed as the mean ± S.E.M., where *a* indicates $p < 0.05$ in comparison with control group, according to ANOVA, followed by Tukey's post-test.

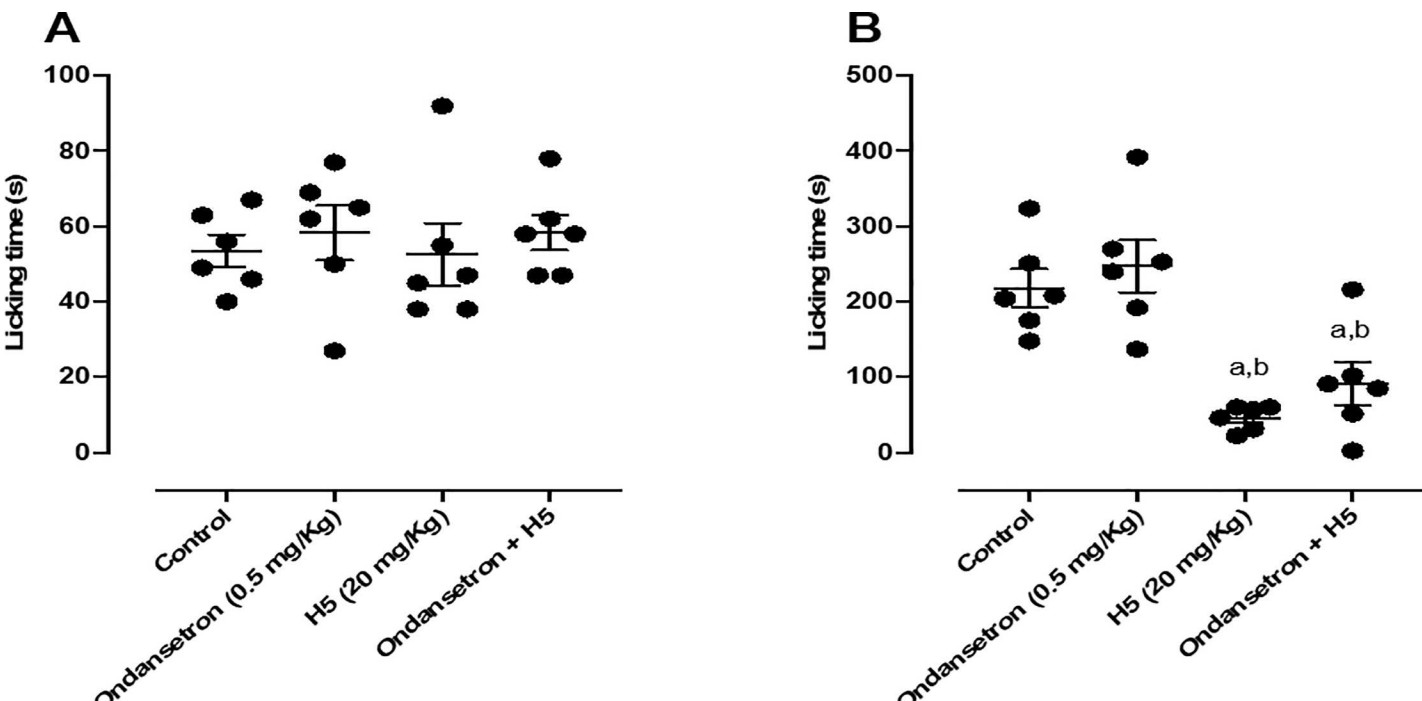

**Fig 7.** Effect of ondansetron (0.5 mg/kg, i.p.), H5 (20 mg/kg, p.o.), ondansetron + H5 in the first (A) and second (B) phases of the formalin-induced nociception test in mice (n = 6, per group). Values are expressed as the mean ± S.E.M., where *a* indicates $p < 0.05$ in comparison with control group and *b* indicates $p < 0.05$ in comparison with the ondansetron group, according to ANOVA, followed by Tukey's post-test.

5-HT$_1$ receptors are implicated in the process of antinociception, whereas 5-HT$_2$ receptors have pronociceptive effects [58–60].

When animals were pretreated with atropine (0.1 mg/kg, i.p.), the pharmacological effect of H5 (20 mg/kg, p.o.) was not reversed in the second phase of the test (Fig 8), suggesting that its peripheral antinociceptive response was not involved with the muscarinic system.

When animals were pretreated with glibenclamide (2 mg/kg, i.p.), the pharmacological effect of H5 (20 mg/kg, p.o.) was not reversed in the second phase of the test (Fig 9), suggesting that its peripheral antinociceptive response was not involved with the ATP sensitive potassium channels.

When animals were pretreated with ruthenium red (3 mg/kg, i.p.), the pharmacological effect of H5 (20 mg/kg, p.o.) was not reversed in the second phase of the test (Fig 10), suggesting that its peripheral antinociceptive response was not involved with the vanilloid system.

We assessed the anti-inflammatory potential of H5. Firstly, the anti-inflammatory effect of H5 was assessed through acute inflammation tests, such as the leukocyte migration in the carrageenan-induced peritoneal cavity test. In this model, H5 reduced leukocyte migration independent of dose (Fig 11). The anti-inflammatory effect of H5 (20mg/kg—37.17% and 40 mg/kg—47.42%) was equivalent to that observed for dexamethasone (2 mg/kg—57.34%).

Another methodology used was the carrageenan-induced hind paw edema model. In this test, H5 significantly decreased ($p < 0.05$) paw edema at all tested doses, especially at 1, 2, 3, and 4 hours after hydrazone treatment, suggesting a pronounced anti-inflammatory effect as shown in Fig 12.

In the carrageenan-induced paw edema model, stimulated inflammation promotes the release of inflammatory mediators in two phases. The first phase occurs one hour after the administration of carrageenan. Then, histamine, serotonin, and cytokines are released.

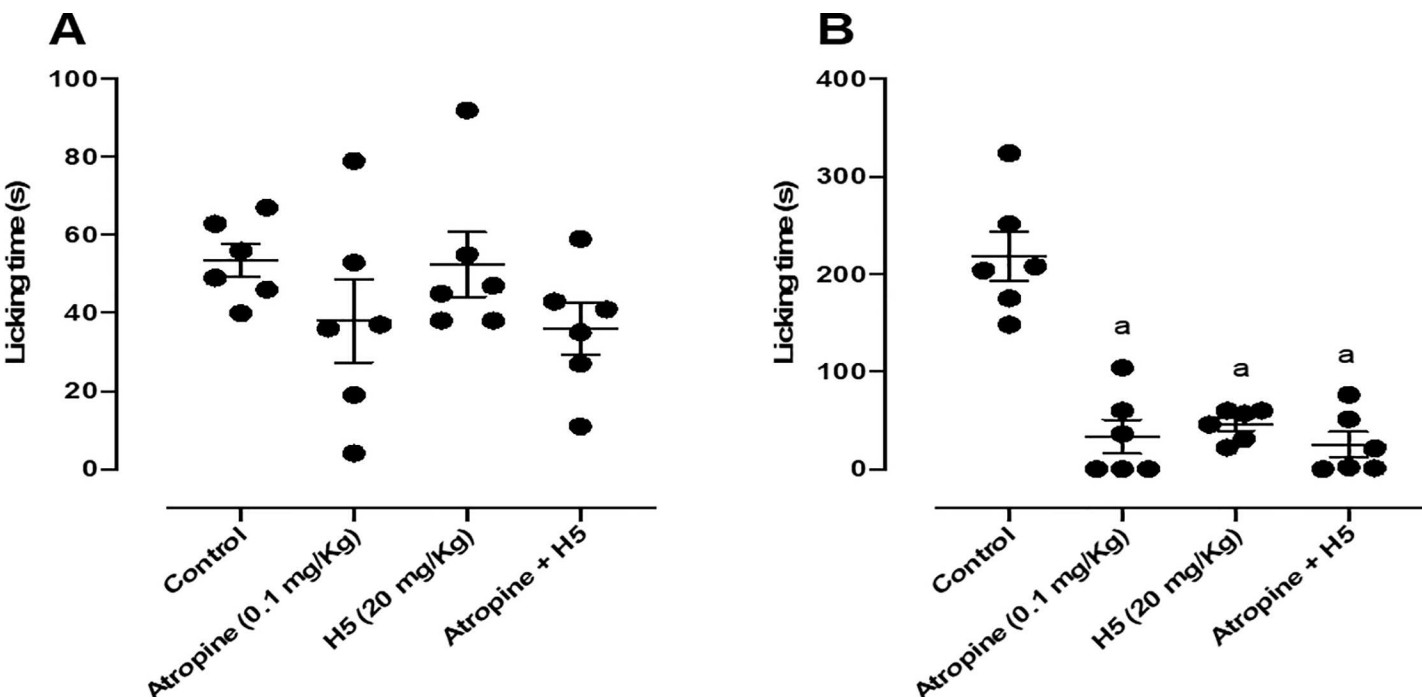

**Fig 8.** Effect of atropine (0.1 mg/kg, i.p.), H5 (20 mg/kg, p.o.), atropine + H5 in the first (A) and second (B) phases of the formalin-induced nociception test in mice (n = 6, per group). Values are expressed as the mean ± S.E.M., where *a* indicates $p < 0.05$ in comparison with control group, according to ANOVA, followed by Tukey's post-test.

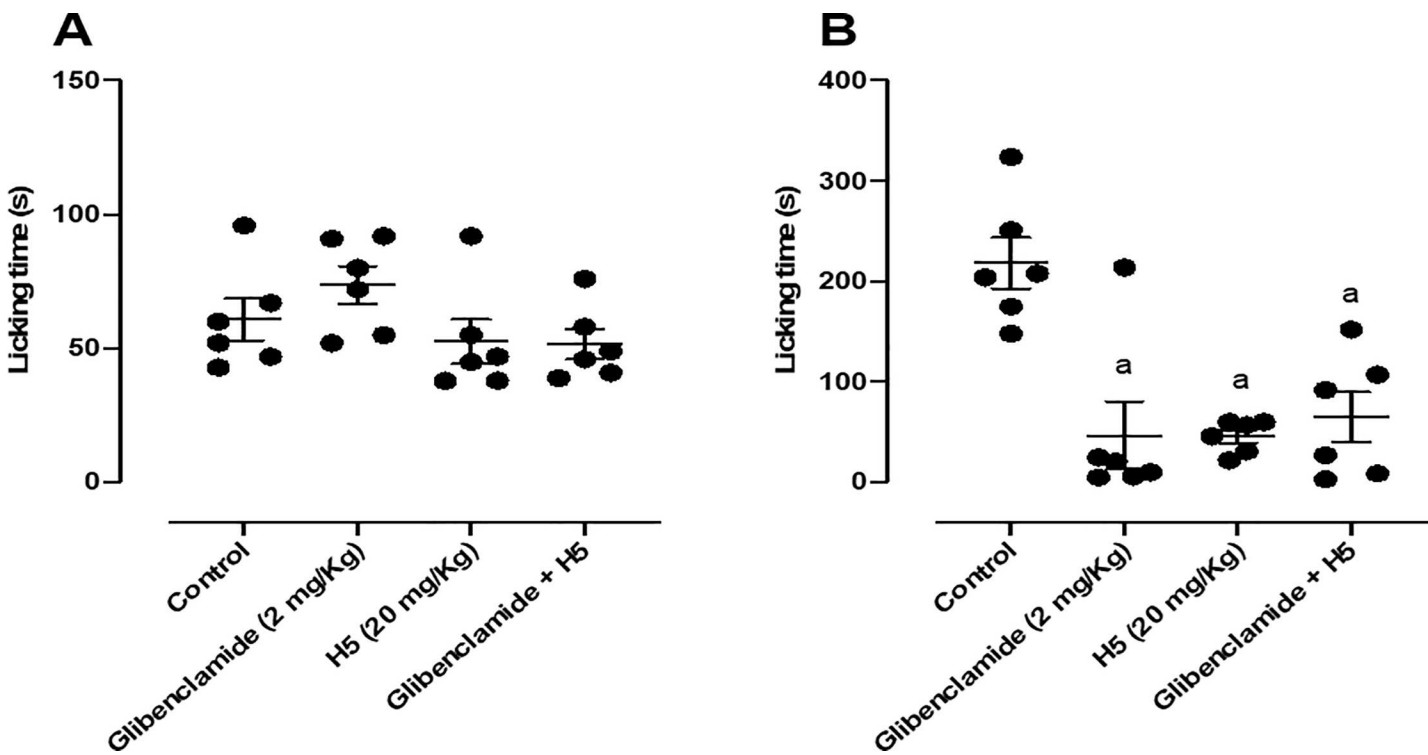

**Fig 9.** Effect of glibenclamide (2 mg/kg, i.p.), H5 (20 mg/kg, p.o.), glibenclamide + H5 in the first (A) and second (B) phases of the formalin-induced nociception test in mice (n = 6, per group). Values are expressed as the mean ± S.E.M., where *a* indicates $p < 0.05$ in comparison with control group, according to ANOVA, followed by Tukey's post-test.

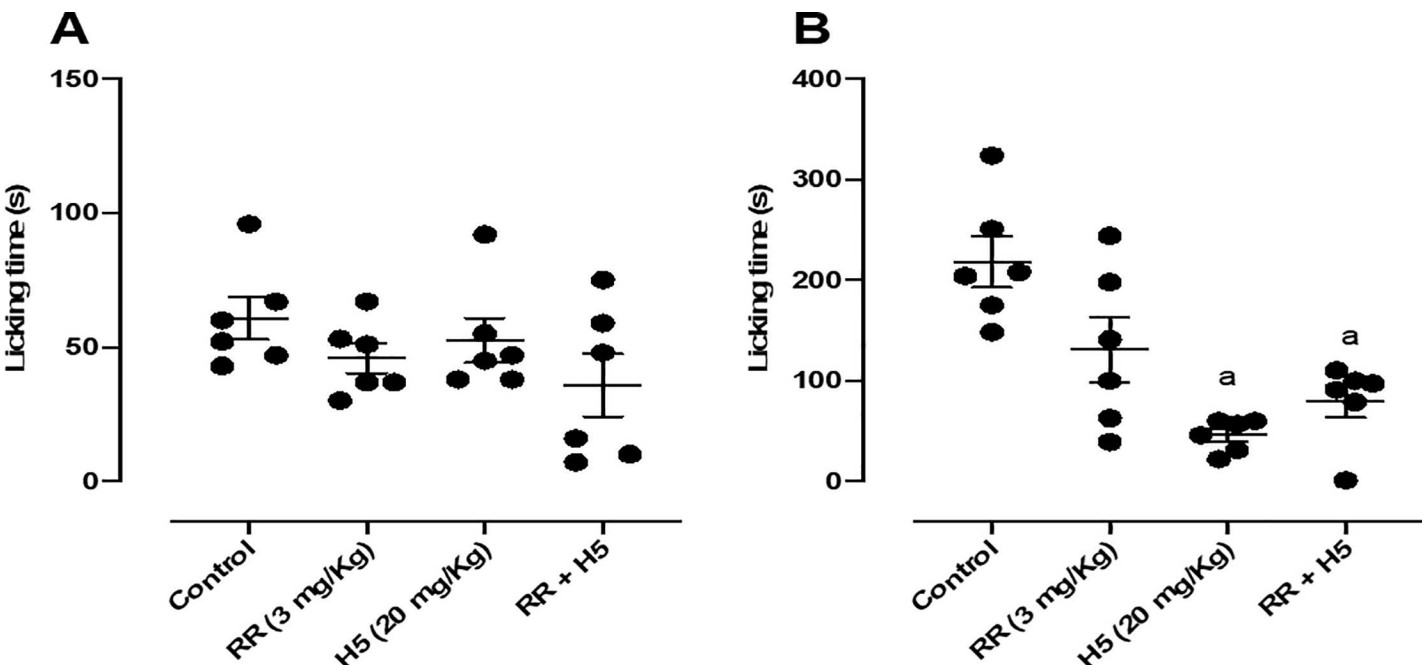

**Fig 10.** Effect of ruthenium red (3 mg/kg, i.p.), H5 (20 mg/kg, p.o.), ruthenium red + H5 in the first (A) and second (B) phases of the formalin-induced nociception test in mice (n = 6, per group). Values are expressed as the mean ± S.E.M., where *a* indicates $p < 0.05$ in comparison with control group, according to ANOVA, followed by Tukey's post-test.

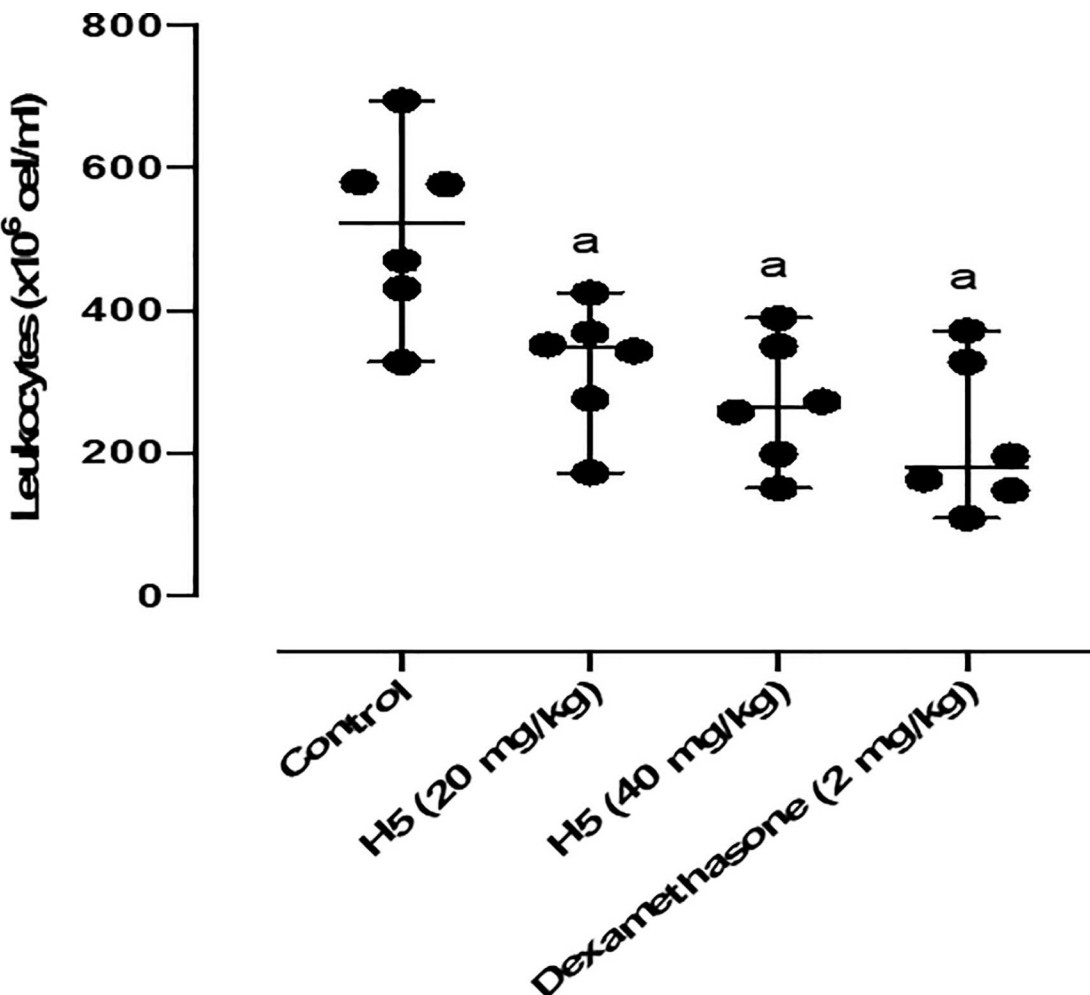

**Fig 11. Effect of H5 (20 and 40 mg/kg, p.o.) and dexamethasone (2 mg/kg, i.p.) on leukocyte migration into the peritoneal cavity induced by carrageenan in mice.** Values are expressed as the mean ± S.E.M. (n = 6, per group), where *a* indicates $p < 0.05$, significantly different from the control group, according to ANOVA, followed by Tukey's test.

Meanwhile, the second phase is characterized by the release of bradykinins, proteases, and prostaglandins, for example. Therefore, the second phase is more sensitive to clinically used anti-inflammatory drugs (ex: diclofenac), which promote the inhibition of cycloxygenases (COX-1 and COX-2), inhibiting the synthesis of prostaglandins [61].

As histamine is one of the first mediators produced (first phase), its vasodilator action is essential for edema formation [14, 62]. Therefore, a similar protocol was performed using histamine to induce paw edema in order to assess the involvement of histaminergic receptors. Fig 13 shows that H5 (20 mg/kg, p.o.) significantly reduced ($p < 0.05$) histamine-induced paw edema at 30, 60, 90, 120, and 150 minutes, suggesting a possible involvement of histamine receptors in its anti-inflammatory effect.

In addition, the rota-rod test was performed to assess the influence of H5 on motor coordination. Rota-rod test showed that there was a change in motor coordination of animals treated with H5 at all doses after 1, 1.5, and 2 h of the administration of H5 (Fig 14). Similarly, diazepam significantly decreased the permanence time on the bar when compared to the negative control.

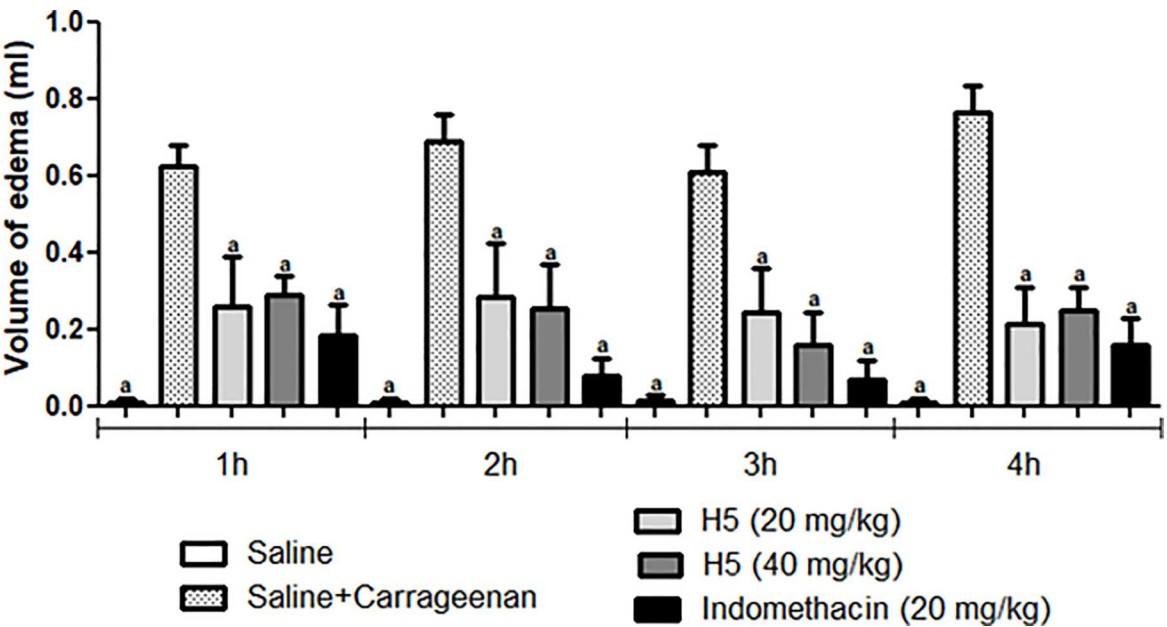

**Fig 12. Effect of H5 (20 and 40 mg/kg, p.o.) and indomethacin (20 mg/kg, i.p.) on paw edema induced by carrageenan in mice.** The sham group was treated only with saline, whereas the control group received saline and carrageenan. Values are expressed as the mean ± S.E.M. (n = 6, per group), where *a* indicates $p < 0.05$, significantly different from the control group, according to ANOVA, followed by Tukey's post-test.

H5 toxicity was assessed using the *Artemia salina* test. The *Artemia salina* assay is a simple, economical, and efficient method for determining acute toxicity. Table 1 describes the lethality rates of H5, positive control (paracetamol, 800 μg/ml), and negative control (saline). It was observed that the positive control showed 40% lethality in the first 24 hours and after 48 hours

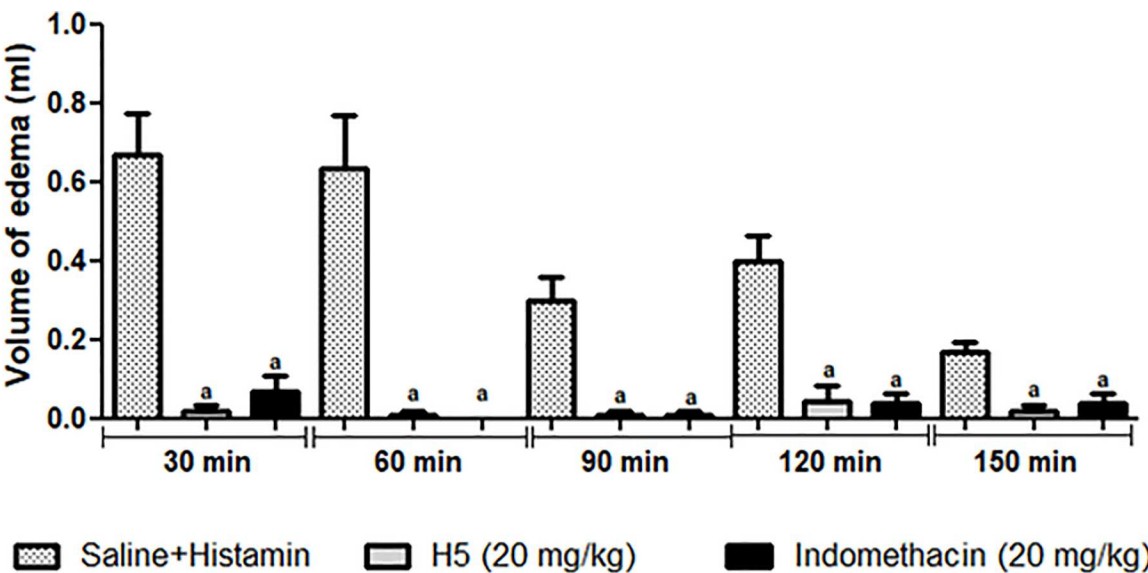

**Fig 13. Effect of H5 (20 mg/kg, p.o.) on paw edema induced by histamine in mice.** The sham group was treated only with saline, whereas the control group received saline and histamine. Values are expressed as the mean ± S.E.M. (n = 6, per group), where *a* indicates $p < 0.05$, significantly different from the control group, according to ANOVA, followed by Tukey's post-test.

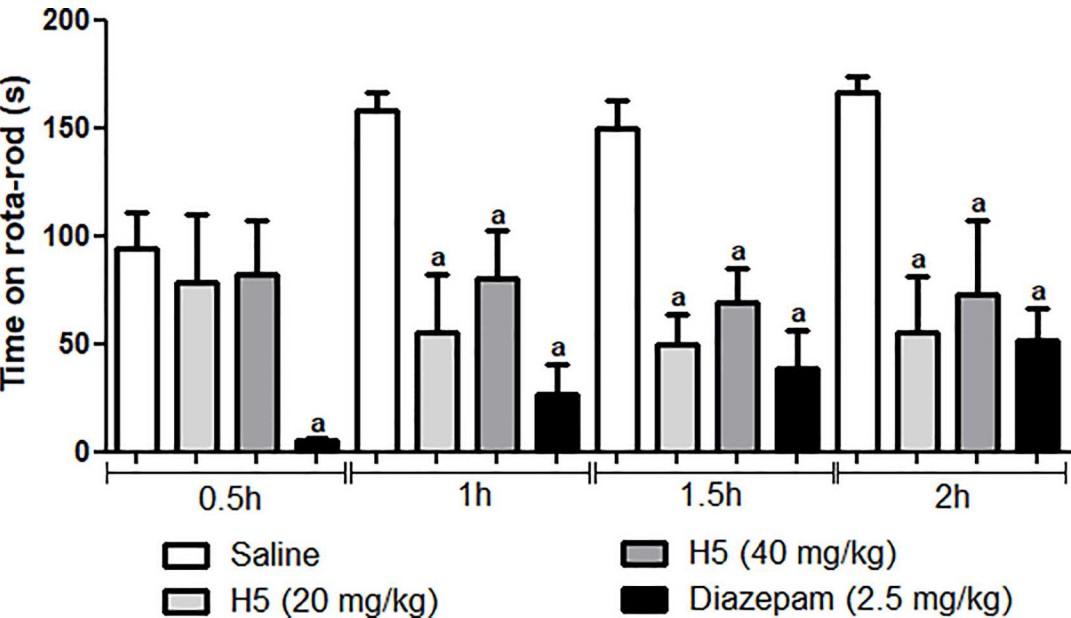

**Fig 14. Effects of H5 (20 and 40 mg/kg, p.o.), and diazepam (2.5 mg/kg, i.p.) in the Rota-rod test in mice.** Values are expressed as mean ± S.E.M. (n = 6, per group), where *a* indicates $p < 0.05$, significantly different from the control group, according to ANOVA, followed by Tukey's post-test.

it resulted in 100%. In the negative control, only one larva of the triplicates did not survive, which showed a lethality rate of 3.4% after 48 hours. For H5 in the first 24 hours, a lethality rate of 54% was observed at a concentration of 250 μg/ml, and after 48 hours a higher lethality rate was observed from the concentration of 100 μg/ml.

According to the literature, the cytotoxic activity against *A. salina* was considered weak when the $LC_{50}$ was between 500 and 1000 μg/ml, moderate when the $LC_{50}$ was between 100 and 500 μg/ml, as strong when the $LC_{50}$ ranged from 0 to 100 μg/ml. In this sense, the results indicated that H5 present moderate toxicity after 24 hours and strong toxicity after 48 hours since it presented $LC_{50}$ of 210.6 μg/ml and 81.95 μg/ml, respectively (Table 2) [32, 63].

Because H5 is an antinociceptive and anti-inflammatory drug candidate, we performed a docking study to analyze its interaction with the COX-2 enzyme. It is known that pharmacological inhibition of COX-2 can relieve inflammation and pain symptoms.

**Table 1. Lethality rate of *Artemia salina* nauplius to hydrazone (H5).**

| Sample | Concentration (μg/ml) | Lethality rate– 24h (%) | Lethality rate– 48h (%) |
|---|---|---|---|
| H5 | 1 | 0 | 4 |
| | 50 | 7 | 34 |
| | 100 | 17 | 57 |
| | 250 | 54 | 84 |
| | 500 | 100 | 100 |
| | 1000 | 100 | 100 |
| CP | 800 | 40 | 100 |
| CN | 0.0038 | 0 | 3.4 |

**Table 2. *Artemia salina* toxicity test of hydrazone H5.**

| Sample | $LC_{50} \pm SD$ of H5 (µg/ml) |
|---|---|
| 24 hours | 210.6 ± 68.38 |
| 48 hours | 81.95 ± 11.10 |

For docking experiments, we performed the redocking of meloxicam (MXM) complexed with X-ray crystallographic structure of murine COX-2 enzyme (PDB ID 4M11) in order to validate our methodology. We got the best RMSD value of 0.28 (Fig 15A) for ChemPLP function with a score value of 65.25. Furthermore, we observed that compound H5 presented a higher score (70.54) when compared to meloxicam, being its interaction modes shown in Fig 15B.

H5 and meloxicam fit in the binding site in a similar fashion. However, H5's phthalazine moiety occupies a larger space than the methylthiazole moiety of meloxicam. We can observe a π-π interaction between the methoxyphenyl group and Tyr355 and between phthalazine and Trp387. It seems that the most important interaction is the hydrogen bond that phthalazine has with Tyr355 and Ser530 (Fig 15A). All this interaction through docking studies may explain why H5 has a notorious antinociceptive effect.

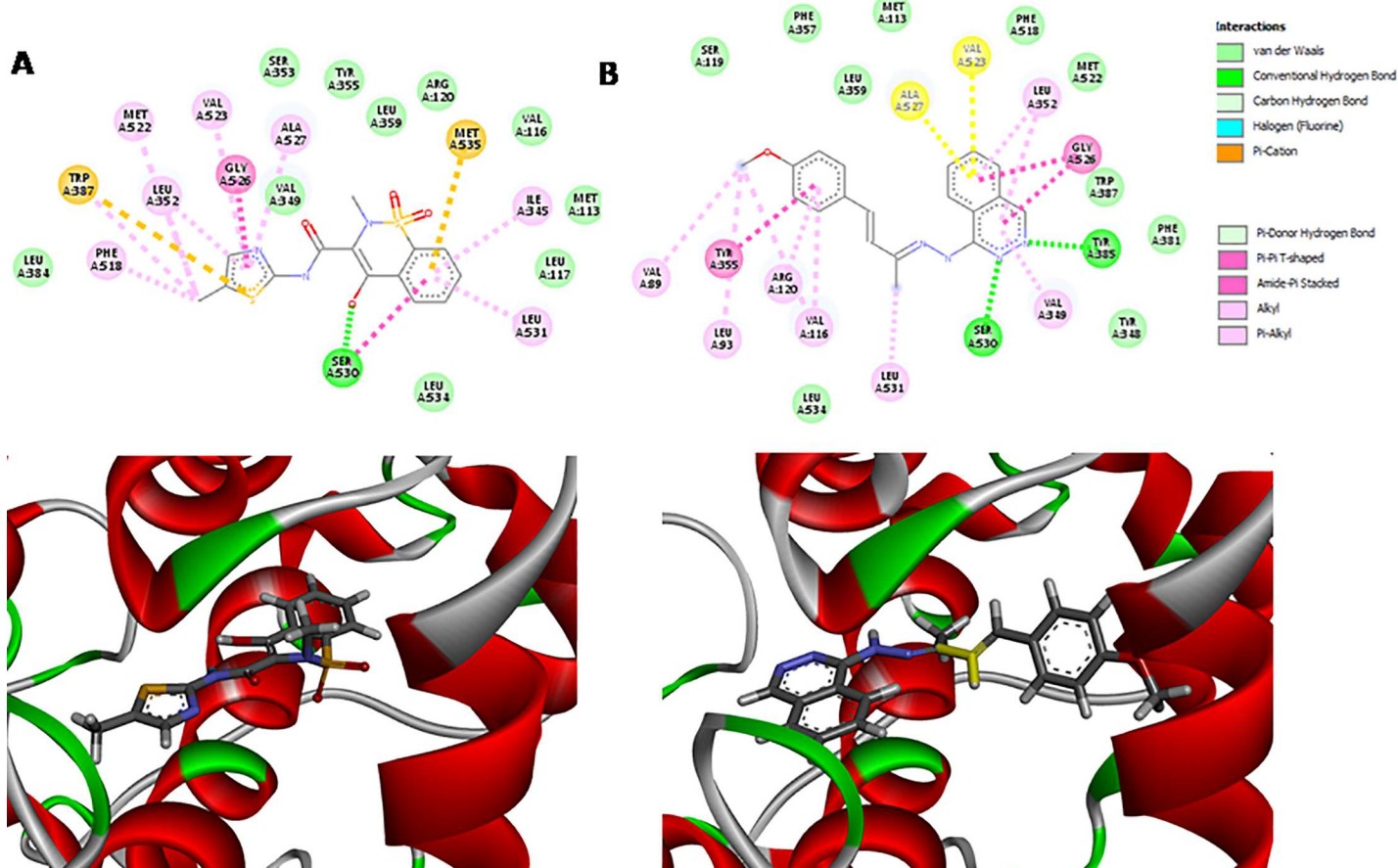

**Fig 15.** Results of the docking procedures for H5 and Meloxicam, (A) Interaction profile of H5 in the murine COX-2 enzyme binding site after the docking study; (B) Interaction profile Meloxicam in the murine COX-2 enzyme binding site after the redocking study.

**Table 3. Comparative _in silico_ physicochemical properties and ADMET profile of the anti-inflammatory drugs indomethacin and meloxicam and new hydrazone series H1 to H5.**

| Predicted Properties* | Compounds | | | | | | |
|---|---|---|---|---|---|---|---|
| | H1 | H2 | H3 | H4 | H5 | Indomethacin | Meloxicam |
| MW (g/mol) | 304.35 | 334.37 | 331.41 | 304.35 | 318.37 | 357.79 | 351.40 |
| H-Donors | 2 | 2 | 1 | 2 | 1 | 1 | 2 |
| H-Acceptors | 5 | 6 | 5 | 5 | 5 | 5 | 7 |
| RotableBonds | 4 | 5 | 5 | 4 | 5 | 4 | 2 |
| TPSA | 70.4 | 79.63 | 53.41 | 70.4 | 59.4 | 68.53 | 136.22 |
| LogP | 3.7 | 2.92 | 4.19 | 3.15 | 3.63 | 4.02 | 2.38 |
| Solubility (mg/ml) | 0.01 | 0.03 | 0.004 | 0.03 | 0.008 | 2.5 | 5.96 |
| Caco-2 (cm/s) | Pe = $215\times10^{-6}$ | Pe = $194\times10^{-6}$ | Pe = $232\times10^{-6}$ | Pe = $205\times10^{-6}$ | Pe = $235\times10^{-6}$ | Pe = $129\times10^{-6}$ | Pe = $233\times10^{-6}$ |
| HIA | 100% | 100% | 100% | 100% | 100% | 100% | 100% |
| F (oral) | 90% | 93% | 80% | 97% | 92% | 99% | 96% |
| PPB | 99% | 99% | 99% | 99% | 99% | 99% | 99% |
| CNS Score | -3.19 | -3.4 | -3.36 | -3.33 | -3.19 | -4.32 | -5.24 |
| HLM | 0.51 | 0.53 | 0.54 | 0.56 | 0.54 | 0.26 | 0.32 |
| hERG | 0.48 | 0.47 | 0.49 | 0.48 | 0.48 | 0.23 | 0.41 |
| AMES | 0.52 | 0.53 | 0.53 | 0.52 | 0.52 | 0.27 | 0.21 |

*Determined _in silico_ using the ACD/Percepta Program. MW = molecular weight; H-Donors = hydrogen bond-donors; H-Acceptors = hydrogen bond-acceptors; TPSA = topological polar surface area; LogP = the logarithm of the drug partition coefficient between n-octanol and water; Caco-2 = human epithelial cell line Caco-2; HIA = human intestinal absorption; F = Bioavailability; CNS = central nervous system; HLM = human liver microsomes; hERG = the human Ether-à-go-go-Related Gene; AMES = Ames test = _Salmonella typhimurium_ reverse mutation assay.

According to the physicochemical characteristics and ADMET profile shown in Table 3, H5 did not violate Lipinsky's rule of five (Ro5), evidencing that this compound has properties that would make it a likely orally active drug in humans [14, 64]. However, it has been predicted that H5 has a lower solubility in comparison with the anti-inflammatory drugs indomethacin and meloxicam. Regarding the comparative ADMET profile of the cited drugs, including H5, it has been predicted that they are highly absorbed (HIA = 100%), highly permeable (Pe > $7\times10^{-6}$ cm/s) and extensively bound to plasma protein (PPB > 90%). _In silico_ analysis of the three compounds has indicated a great oral bioavailability (F = 80 to 99%). The main discrepancies among H5, indomethacin, and meloxicam rely on two specific properties: the metabolic stability in human liver microsomes (HLM) and the ability to access the CNS. In this sense, indomethacin and meloxicam were predicted to have stability in HLM (scores of 0.26 and 0.32, respectively), while H5 presented an undefined result (score = 0.54). Both reference drugs have been predicted as non-penetrant on the CNS (scores of -4.32 and -5.24, respectively) whereas H5 had this ability. Regarding H5 toxicity, we assessed the ability of this compound to inhibit hERG (the human Ether-à-go-go-Related Gene), so we could predict its mutagenic profile (i.e. probability of a positive Ames test). The outcomes were then converted into classification scores and showed that H5 has undefined hERG and mutagenic activities (score > 0.33 and ≤ 0.67). Although _in silico_ approach was unable to predict the toxicological profile of H5, this very analysis suggests an adequate pharmacokinetic profile for this agent [14].

## Conclusion

Given the results presented here, H1, H2, H3, H4, and H5 showed prominent anti-nociceptive and anti-inflammatory effects in both experimental models tested. Among the five hydrazone

derivatives tested, H5 was significantly more active concerning antinociceptive and anti-inflammatory activities in all experimental models. Its antinociceptive mechanism of action appears to be peripheral, with the involvement of the opioid signaling pathway. Furthermore, the anti-inflammatory effect of H5 may be involved with the histaminergic receptor pathways. In addition to that, H5 promotes COX-2 inhibition, as demonstrated by the molecular docking study. Regarding the *in silico* studies, H5 presented an adequate pharmacokinetic profile. In short, H5 has emerged as a strong candidate for an antinociceptive and multi-target anti-inflammatory.

## Supporting information

**S1 File.**
(DOCX)

**S1 Dataset. Raw data.**
(RAR)

## Author Contributions

**Conceptualization:** Maria Alice Miranda Bezerra Medeiros, Mariana Gama e Silva, Jackson de Menezes Barbosa, Érica Martins de Lavor, Tiago Feitosa Ribeiro, Cícero André Ferreira Macedo, Luiz Antonio Miranda de Souza Duarte-Filho, Thiala Alves Feitosa, Jussara de Jesus Silva, Harold Hilarion Fokoue, Cleônia Roberta Melo Araújo, Arlan de Assis Gonsalves, Luciano Augusto de Araújo Ribeiro, Jackson Roberto Guedes da Silva Almeida.

**Methodology:** Maria Alice Miranda Bezerra Medeiros, Mariana Gama e Silva, Jackson de Menezes Barbosa, Érica Martins de Lavor, Tiago Feitosa Ribeiro, Cícero André Ferreira Macedo, Luiz Antonio Miranda de Souza Duarte-Filho, Thiala Alves Feitosa, Jussara de Jesus Silva, Harold Hilarion Fokoue, Cleônia Roberta Melo Araújo, Arlan de Assis Gonsalves, Luciano Augusto de Araújo Ribeiro, Jackson Roberto Guedes da Silva Almeida.

**Writing – original draft:** Maria Alice Miranda Bezerra Medeiros, Mariana Gama e Silva, Jackson de Menezes Barbosa, Érica Martins de Lavor, Tiago Feitosa Ribeiro, Cícero André Ferreira Macedo, Luiz Antonio Miranda de Souza Duarte-Filho, Thiala Alves Feitosa, Jussara de Jesus Silva, Harold Hilarion Fokoue, Cleônia Roberta Melo Araújo, Arlan de Assis Gonsalves, Luciano Augusto de Araújo Ribeiro, Jackson Roberto Guedes da Silva Almeida.

**Writing – review & editing:** Maria Alice Miranda Bezerra Medeiros, Mariana Gama e Silva, Jackson de Menezes Barbosa, Érica Martins de Lavor, Tiago Feitosa Ribeiro, Cícero André Ferreira Macedo, Luiz Antonio Miranda de Souza Duarte-Filho, Thiala Alves Feitosa, Jussara de Jesus Silva, Harold Hilarion Fokoue, Cleônia Roberta Melo Araújo, Arlan de Assis Gonsalves, Luciano Augusto de Araújo Ribeiro, Jackson Roberto Guedes da Silva Almeida.

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
