## [Decision Letter · Decision Letter 0]

5 Mar 2021

PONE-D-21-02843

Antinociceptive and anti-inflammatory effects of hydrazone derivatives and their possible mechanism of action in mice

PLOS ONE

Dear Dr. Almeida,

Thank you for submitting your manuscript to PLOS ONE. After careful consideration, we feel that it has merit but does not fully meet PLOS ONE’s publication criteria as it currently stands. Therefore, we invite you to submit a revised version of the manuscript that addresses the points raised during the review process.

In particular, please pay special attention to the comments on figure consolidation and conclusions drawn from the data.  

We look forward to receiving your revised manuscript.

Kind regards,

John M. Streicher, Ph.D.

Academic Editor

PLOS ONE

Journal Requirements:

2) In your Methods section, please provide additional details regarding the animals used in your study and ensure you have described the source. For more information regarding PLOS' policy on materials sharing and reporting, see https://journals.plos.org/plosone/s/materials-and-software-sharing#loc-sharing-materials.

3)  Thank you for submitting the above manuscript to PLOS ONE. During our internal evaluation of the manuscript, we found significant text overlap between your submission and the following previously published work, some of which you are an author.

https://journals.plos.org/plosone/article?id=10.1371%2Fjournal.pone.0199009

Please revise the manuscript to rephrase the duplicated text, cite your sources, and provide details as to how the current manuscript advances on previous work. Please note that further consideration is dependent on the submission of a manuscript that addresses these concerns about the overlap in text with published work.

Reviewers' comments:

Reviewer's Responses to Questions

**Comments to the Author**

1. Is the manuscript technically sound, and do the data support the conclusions?

Reviewer #1: Partly

Reviewer #2: Partly

2. Has the statistical analysis been performed appropriately and rigorously? 

Reviewer #1: Yes

Reviewer #2: Yes

3. Have the authors made all data underlying the findings in their manuscript fully available?

Reviewer #1: Yes

Reviewer #2: No

4. Is the manuscript presented in an intelligible fashion and written in standard English?

Reviewer #1: No

Reviewer #2: No

5. Review Comments to the Author

Reviewer #1: In this manuscript by Medeiros et al, explores the analgesic and anti-inflammatory effect of a series of hydrazone derivatives. The compounds appear to have significant efficacy in multiple mouse models. My main concerns for the manuscript are with the lack of a strong and clear rationale/introduction for the class of molecules being generated, lack of a discussion how the obtained results matter in relation to known drugs, and the interpretation of the ‘mechanistic’ experimental data. As someone that serves on an institutional animal care and use committee, I would personally not have approved the research design, which more or less is a phenotypic screen, but could have been done largely in vitro first, before moving to animals. Yet, this study was approved by PIs IACUC and thus I will not hold this against the authors, but I want to be on record with my personal reservation.

Major comments.

• The rationale for choosing to make hydrazone derivatives and testing them in the animal models is not well described.

• Page 5: animal weight 30 +/- 40 g and humidity 60 +/- 80% must be wrong.

• Please describe the rationale for choosing 20 and 40 mg/kg doses to test.

• The Artemia salina toxicity test needs a more detailed description

• Is the indomethacin and morphine data in figures 1-5 different for each group or is the same data represented multiple times. The latter would be unacceptable without clearly stating so.

• Page 18: Proof needs to be provided for the statement that ‘H5 is more chemically stable’

• The mention of celecoxib and cox-2 on page 19 comes out of nowhere and is only brought up, but not discussed. Similarly on page 21, the mention of bergamot and flavonoids seem irrelevant.

• The rationale for the use of each pharmacological inhibitor is poorly described. For example why was the ondansetron used.

• Given that L-Arg decreases licking time more than H5+ L-Arg, means that no conclusion can be drawn about any relation to L-Arg and H5. Any such conclusion needs to be removed (e.g. on page 22).

• The statement “shows that H5 (20 mg/kg, p.o.) significantly reduced (p<0.05) histamineinduced

• paw edema at 30, 60, 90, 120, and 150 minutes, suggesting the involvement of

• histamine receptors in its anti-inflammatory effect” is too strong and needs to be in line with what the data really allows for.

• Why was the rotarod test performed, one would expect H5 to be compared to a analgesic or anti-inflammatory drug that does perform poorly in this test.

• What were the positive and negative controls for the Artemia salina test.

• Page 31 The sentence “…the substance that has an LC50 value less than 1000 μg/ml compared to Artemia salina.” does not make sense.

• It seems that H5 is more toxic than the positive control, please explain

• Page 32: For validation, redocking studies were performed, comes out of nowhere. Validation of what. Re-docking of what. Why is this molecule docked at COX-2

• Meloxicam is used in the docking, but is not used at all in the in vivo models to make a more logical comparison.

• Page 33: “This may explain the H5 markable antinociceptive effect.” Is way too ambitious of a statement

• Figure 22 and 23 the angle of view provides inferior insight into the binding mode/site.

Minor comments.

• Figures need to be pooled into multi-panel figures to reduce the figure number down from 23 to a more manageable number.

• There are numerous grammatical errors and instances of dubious word choices in the manuscript, too many to list, but it will be valuable to re-read the manuscript and improve where possible. Purely as example: sentence 1 of the abstract “pain and inflammation ..resulting from imminent tissue damage”. Sentence 3 of the abstract starts with “therefore”, but is not really a logical continuation of the prior sentence. Sentence 6 ”greatest potential” should read “greatest potency”.

• Instead of stating “ at the highest doses tested” rephrase as “both tested doses”.

• Fore figures, please use individual dots instead of bar graphs.

Reviewer #2: This was an interesting study on novel compounds to treat pain and inflammation. However, questions and concerns that arose are listed below:

1) What is the reason for administering reference drugs 30 minutes prior to nociceptive agent but test drugs being administered at 1 hour prior to nociceptive agent? Any rationale for this difference in time intervals?

2) Each graph is placed separately as a figure. This makes the manuscript unnecessarily bulky and long. Could the authors please consolidate their findings? For example, Figure 2 for acetic acid test can comprise of graphs for H1 through H5 put together (current figures 2-6) and so on. Also, there are a massive number of supplementary figures, which could again be consolidated.

3) It would be helpful for the authors to insert a table listing all statistical analyses performed with their respective exact p-values. Right now, significance is only indicated as p<0.05 in figure legends.

4) It is unclear if an unbiased video recording and analysis software was used to document pain behavior after administering the drugs. If behaviors were recorded manually, what was done to avoid experimenter error and bias?

5) The authors did a good job of explaining the rationale and action of already established anti-nociceptive drugs used as controls. However, they missed explanations for specific actions for a few of these namely, Ondansetron, atropine, glibenclamide, etc.

The authors should cross-check whether a separate, more elaborate discussion section is required by the journal. The manuscript needs substantial revision in addition to addressing the comments above if this is the case as no discussion and speculation has been provided. Also, the limitation that H5 is CNS-penetrant has been glossed over. This can affect many aspects of central sensitization during pain and inflammation, and it should be discussed more.

6. PLOS authors have the option to publish the peer review history of their article (what does this mean?). If published, this will include your full peer review and any attached files.

Reviewer #1: No

Reviewer #2: No

---

## [Decision Letter · Decision Letter 1]

16 Jun 2021

PONE-D-21-02843R1

Antinociceptive and anti-inflammatory effects of hydrazone derivatives and their possible mechanism of action in mice

PLOS ONE

Dear Dr. Almeida,

Thank you for submitting your manuscript to PLOS ONE. After careful consideration, we feel that it has merit but does not fully meet PLOS ONE’s publication criteria as it currently stands. Therefore, we invite you to submit a revised version of the manuscript that addresses the points raised during the review process.

We look forward to receiving your revised manuscript.

Kind regards,

John M. Streicher, Ph.D.

Academic Editor

PLOS ONE

Journal Requirements:

Additional Editor Comments (if provided):

Thank you for your revision. Please address the minor concerns by Reviewer 1; I can evaluate your subsequent revision without sending out for another round of review.

Reviewers' comments:

Reviewer's Responses to Questions

**Comments to the Author**

1. If the authors have adequately addressed your comments raised in a previous round of review and you feel that this manuscript is now acceptable for publication, you may indicate that here to bypass the “Comments to the Author” section, enter your conflict of interest statement in the “Confidential to Editor” section, and submit your "Accept" recommendation.

Reviewer #1: (No Response)

Reviewer #2: All comments have been addressed

2. Is the manuscript technically sound, and do the data support the conclusions?

Reviewer #1: Partly

Reviewer #2: Yes

3. Has the statistical analysis been performed appropriately and rigorously? 

Reviewer #1: Yes

Reviewer #2: Yes

4. Have the authors made all data underlying the findings in their manuscript fully available?

Reviewer #1: Yes

Reviewer #2: Yes

5. Is the manuscript presented in an intelligible fashion and written in standard English?

Reviewer #1: No

Reviewer #2: Yes

6. Review Comments to the Author

Reviewer #1: The authors addressed several of my prior concerns, but several were not addressed. It is critical that in figures 2-4 it Is made clear that the indomethacin and morphine data are identical in each panel.

Major/critical concerns:

Figure 2-4: You have to explicitly state that in each panel the indomethacin and morphine are identical/copy pasted. Not doing so gives the false and unethical impression that you ran a new positive control each time.

The statement “The anti-inflammatory effect of H5 appears to involve histaminergic receptors” is not based on the data. Just because the molecule can reduce inflammation induced by histamine does not mean it acts through histaminergic receptors. As a simple analogy: If morphine reduces pain from gunshot wounds, it doesn’t mean morphine acts on guns.

It seems that H5 is more toxic than the positive control, please explain

Minor concerns/edits

Please remove that pain results from imminent tissue damage from the 1st sentence of the abstract and introduction.

The sentence “Such panorama indicates that the application of classical in vivo animal experiments has an important role in drug discovery” is not very clear.

What is the point of the statement “Additionally, a variety of hydrazone derivatives have been developed to minimize gastrointestinal discomfort and toxicity, especially when it comes to analgesic drugs” minimizing GI discomfort and toxicity doesn’t really link to the current study where the antinociceptive and anti-inflammatory of the derivatives are being assessed.

Page 15 “When animals were pretreated with naloxone (1.5 mg/kg, i.p.), the pharmacological

effect of H5 (20 mg/kg, p.o.) was completely reversed in the second phase of the test (Fig 5),

suggesting that its peripheral antinociceptive response was involved at least in part with the

opioid system.” Is repeated on page 16-17 “When animals were pretreated with naloxone, the pharmacological effect of H5 was completely reversed in the second phase of the formalin test, suggesting that its peripheral antinociceptive response is involved, at least in part, with the opioid system”

Figure 15 the two interaction map is low resolution and hard to read. The docked molecule is very planar and the angle of the docked ligand hides a lot of the molecule, a small tilt would show much more of the ligand

For figures, please use individual dots for each datapoint instead of bar graphs.

Reviewer #2: (No Response)

7. PLOS authors have the option to publish the peer review history of their article (what does this mean?). If published, this will include your full peer review and any attached files.

Reviewer #1: No

Reviewer #2: No

---

## [Author Response · Author response to Decision Letter 1]

29 Jul 2021

Response to Reviewers

* The authors addressed several of my prior concerns, but several were not addressed. It is critical that in figures 2-4 it Is made clear that the indomethacin and morphine data are identical in each panel.

Major/critical concerns:

Figure 2-4: You have to explicitly state that in each panel the indomethacin and morphine are identical/copy pasted. Not doing so gives the false and unethical impression that you ran a new positive control each time.

Thank you for your observations. The positive controls used in the graphs of figure 2 were the same, as the experiments were carried out on the same day, the animals used had the same age and weight range, in addition to being exposed to the same environmental conditions.

The same occurred for figures 3 and 4.

* The statement “The anti-inflammatory effect of H5 appears to involve histaminergic receptors” is not based on the data. Just because the molecule can reduce inflammation induced by histamine does not mean it acts through histaminergic receptors. As a simple analogy: If morphine reduces pain from gunshot wounds, it doesn’t mean morphine acts on guns.

Thank you for this consideration. I guess we might have a semantic problem in the referred sentence. Perhaps the word “appears” sounded more like a certainty than a possibility, but it is not what we meant. In this sense, the text has been improved to fix misinterpretations about what we judge as possible participation of histaminergic receptors in the anti-inflammatory effect of H5. 

* It seems that H5 is more toxic than the positive control, please explain.

Regarding Table 1, a similar result of H5 was observed in relation to the positive control at higher concentrations. The results indicated that H5 present moderate toxicity after 24 hours and strong toxicity after 48 hours since it presented LC50 of 210.6 µg/ml and 81.95 µg/ml, respectively (Table 2). 

* Please remove that pain results from imminent tissue damage from the 1st sentence of the abstract and introduction.

We have proceeded with this modification. Thank you.

* The sentence “Such panorama indicates that the application of classical in vivo animal experiments has an important role in drug discovery” is not very clear.

Thank you for this consideration. We have agreed with the reviewer and proceeded with some modifications.

* What is the point of the statement “Additionally, a variety of hydrazone derivatives have been developed to minimize gastrointestinal discomfort and toxicity, especially when it comes to analgesic drugs” minimizing GI discomfort and toxicity doesn’t really link to the current study where the antinociceptive and anti-inflammatory of the derivatives are being assessed.

Thank you for your comment. GI discomfort and toxicity are disadvantages that greatly limit the use of analgesics and anti-inflammatory drugs. This phrase was added to emphasize an advantage of hydrazones over classical anti-inflammatories. However, it deserves some improvement.

* Page 15 “When animals were pretreated with naloxone (1.5 mg/kg, i.p.), the pharmacological effect of H5 (20 mg/kg, p.o.) was completely reversed in the second phase of the test (Fig 5), suggesting that its peripheral antinociceptive response was involved at least in part with the opioid system.” Is repeated on page 16-17 “When animals were pretreated with naloxone, the pharmacological effect of H5 was completely reversed in the second phase of the formalin test, suggesting that its peripheral antinociceptive response is involved, at least in part, with the opioid system”.

Thank you for this observation. We have proceeded with the correction. 

* Figure 15 the two interaction map is low resolution and hard to read. The docked molecule is very planar and the angle of the docked ligand hides a lot of the molecule, a small tilt would show much more of the ligand.

We apologize for that. A high-resolution image replaced figure 15.

* For figures, please use individual dots for each datapoint instead of bar graphs.

Thank you for this consideration. We have agreed with the reviewer and proceeded with some modifications. Figures 12, 13 and 14 were not self-explanatory in the requested model, so we left it in the previous model.

---

## [Editor Report · Decision Letter 2]

21 Sep 2021

Antinociceptive and anti-inflammatory effects of hydrazone derivatives and their possible mechanism of action in mice

PONE-D-21-02843R2

Dear Dr. Almeida,

We’re pleased to inform you that your manuscript has been judged scientifically suitable for publication and will be formally accepted for publication once it meets all outstanding technical requirements.

Kind regards,

John M. Streicher, Ph.D.

Academic Editor

PLOS ONE
---

## [Editor Report · Acceptance letter]

15 Nov 2021

PONE-D-21-02843R2 

Antinociceptive and anti-inflammatory effects of hydrazone derivatives and their possible mechanism of action in mice 

Dear Dr. Almeida:

I'm pleased to inform you that your manuscript has been deemed suitable for publication in PLOS ONE. Congratulations! Your manuscript is now with our production department. 

Kind regards, 

on behalf of

Dr. John M. Streicher 

Academic Editor

PLOS ONE